# Multifunctional and Smart Wound Dressings—A Review on Recent Research Advancements in Skin Regenerative Medicine

**DOI:** 10.3390/pharmaceutics14081574

**Published:** 2022-07-28

**Authors:** Nithya Rani Raju, Ekaterina Silina, Victor Stupin, Natalia Manturova, Saravana Babu Chidambaram, Raghu Ram Achar

**Affiliations:** 1Division of Biochemistry, School of Life Sciences, JSS Academy of Higher Education & Research, Mysuru 570015, Karnataka, India; nithyaranir@jssuni.edu.in; 2Institute of Biodesign and Modeling of Complex Systems, I.M. Sechenov First Moscow State Medical University (Sechenov University), Trubetskaya Street 8, 119991 Moscow, Russia; silinaekaterina@mail.ru; 3Department of Hospital Surgery No 1, N.I. Pirogov Russian National Research Medical University (RNRMU), Ostrovityanova Street 1, 117997 Moscow, Russia; stvictor@bk.ru; 4Department of Plastic and Reconstructive Surgery, Cosmetology and Cell Technologies, N.I. Pirogov Russian National Research Medical University, Ostrovityanova Street 1, 117997 Moscow, Russia; manturovanatali@yandex.ru; 5Department of Pharmacology, JSS College of Pharmacy, JSS Academy of Higher Education & Research, Mysuru 570015, Karnataka, India; saravanababu.c@jssuni.edu.in; 6Centre for Experimental Pharmacology and Toxicology (CPT), Central Animal Facility, JSS Academy of Higher Education & Research, Mysuru 570015, Karnataka, India

**Keywords:** smart wound dressings, multifunctional dressings, biomaterials, nanoparticles, tissue regeneration

## Abstract

The healing of wounds is a dynamic function that necessitates coordination among multiple cell types and an optimal extracellular milieu. Much of the research focused on finding new techniques to improve and manage dermal injuries, chronic injuries, burn injuries, and sepsis, which are frequent medical concerns. A new research strategy involves developing multifunctional dressings to aid innate healing and combat numerous issues that trouble incompletely healed injuries, such as extreme inflammation, ischemic damage, scarring, and wound infection. Natural origin-based compounds offer distinct characteristics, such as excellent biocompatibility, cost-effectiveness, and low toxicity. Researchers have developed biopolymer-based wound dressings with drugs, biomacromolecules, and cells that are cytocompatible, hemostatic, initiate skin rejuvenation and rapid healing, and possess anti-inflammatory and antimicrobial activity. The main goal would be to mimic characteristics of fetal tissue regeneration in the adult healing phase, including complete hair and glandular restoration without delay or scarring. Emerging treatments based on biomaterials, nanoparticles, and biomimetic proteases have the keys to improving wound care and will be a vital addition to the therapeutic toolkit for slow-healing wounds. This study focuses on recent discoveries of several dressings that have undergone extensive pre-clinical development or are now undergoing fundamental research.

## 1. Introduction

Skin is the most important organ in the body, an active immune organ, and the principal barrier between the environment and interior organs [1]. Damages caused for various reasons, such as domestic, sports, and military injuries to the integrity of the skin or organs, are called wounds, which can be mechanical, thermal, chemical, or radiogenic skin trauma. Even though self-healing is one of the characteristics of human skin, the long-term repair process and associated pathogenic consequences such as inflammation and secondary damage, especially for extensive full-thickness wounds, mean that wounds put a high demand on wound dressing design and implementation. Hence, healing is not only a medical issue but also a social and economic concern [2,3,4,5]. In general, the healing process is divided into four phases: hemostasis, inflammation, proliferation, and remodeling [6,7].

The healing of wounds involves interactions between cellular growth factors, components of blood, and extracellular matrix. Cytokines promote healing via a variety of mechanisms, including promoting the formation of basement membrane components, preventing dehydration, increasing inflammation, and promoting the formation of granulation tissue. In adolescents, wound healing take a few days, but in adults, it can take several weeks [8]. Depending on the time required for the stages of healing, wounds may be acute or chronic. In less than four weeks, acute wound repair is devoid of complications and re-establishes all stages of the healing process. Shorter wound closure time is also related to less scar formation. Chronic wounds are deep, full-thickness or partial-thickness injuries that do not heal in less than six weeks. They take a long time to heal and are linked to increased fibrosis, resulting in hypertrophic scars and keloids in some people [9,10]. Chronic wounds have a complicated, inflammatory nature and produce large amounts of exudate, which impedes tissue repair. Hence, these chronic wounds impact a significant percentage of the healthcare system and are likely to shift acute disorders into irreversible systemic damage, culminating in catastrophic effects and even death [11,12]. Ache, limited mobility, discharge, unpleasant smell, a distorted physical image, limited social engagement, economic and community responsibilities, and constraints stemming from the therapy itself are all connected with chronic wounds and limit the everyday routines and living quality of those afflicted [13]. 

## 2. Current Challenges and Perspective

The wound healing process is dependent on several interlinked factors, which are mainly of two types: local and systemic factors (Figure 1) [14,15]. Local factors directly impact the wound characteristics, while systemic factors depend on the individual’s overall status of health or illness. Soreness, sepsis, irradiation, hypothermia and oxygenation are local factors that affect the features of the wound, whereas systemic factors such as malnutrition, age group, gender and diseases affect an individual’s ability to heal [16,17,18]. 

S. Guo and L.A. DiPietro have described in detail how these factors influence each stage by delaying healing process and thus there is a necessity for consideration of these factors during formulating wound care products depending on wound types. Annually, over 305 million people with acute injuries are reported globally. The total count of acute injury observed is over nine times the overall population of cancer patients in the global [18]. Acute wounds include surgical and traumatic wounds, as well as abrasions and superficial burns [19]. Ischemic ulcers, diabetic ulcers, venous ulcers and pressure ulcers are the prevalent chronic damages. Annually, there are reports of 463 million diabetics, chronic foot ulcers affect 9.1 to 26.1 million persons worldwide, and 1% of the world population with venous ulcers. India being one of the topmost countries recording highest trauma injuries, accounts for 10.5 out of 1000 [19]. In the US, it is estimated that annually the expenditure for acute and chronic wound ranges from 28.1 billion to 96.8 billion US dollars [20]. 

In various Ayurvedic scriptures, around 1200 ailments are listed, out of which ninety percent comprise medicinal herbs; five percent contain ores, metallic minerals, and ores; and the remaining five percent include animal and marine products [21]. Alternative and complementary medicines based on natural plant-based bioactive chemicals are in high demand. The process of wound healing hastens in multiple ways by accessing different dressings for an entire variety of wounds through various technological advancements. Choosing the correct component dressing for a specific injury is critical for rapid repair of the injured area [22]. According to a study, the market for wound care products will reach 15 to 22 billion US dollars by 2024, as scientists, engineers, and clinicians are working towards developing technologically advanced smart wound care products [19]. 

## 3. Characteristics of Multifunctional Wound Dressing

A smart and multifunctional dressing material must be applied based on the wound type. The proficiency (a) to establish a damp environment; (b) to facilitate movements in the epidermis; (c) to encourage blood vessel formation and tissue regeneration; (d) to provide good permeability for oxygen and water vapor; (e) to ensure a decent heat in the wound site to stimulate the circulation of blood; (f) to protect from infectious pathogens; (g) to be non-adhesive to the injured area and to be easily removable; (h) to encourage debridement and to promote leucocyte movement and enzyme build-up; (i) sterile, non-toxic, and allergy-free; (j) biocompatible and effective drug administration; (k) high absorption potential for wound exudates should all be considered when choosing an ideal dressing (Figure 2) [23,24,25].

### 3.1. pH Responsive Wound Dressings

pH is one such factor that exerts a major influence on the process of healing in all four stages. The pH of the healthy epidermis is around 4.5–6.5, whereas acute and chronic wounds have their own pH range. Due to microbial infection and production of alkaline by-products, the wound pH may rise to 7–9. Many methods are being used for monitoring pH in the wound healing process, such as colorimetry, electrochemical & electromechanical methods. These methods are laborious, costly, and consume more utility. Thus, the development of sensor-based dressings would be easily monitorable and cost-effective. Flex circuit transducers, optical sensor-based dyes, pH-sensitive color dyes, pH-sensitive electrodes, and carbon dots are some strategies used to manufacture pH-sensitive wound dressings [24,26].

### 3.2. Temperature Responsive Wound Dressings

During wound recovery, temperature is an important parameter that depends on various enzymatic and biochemical reactions occurring in the injured site. Studies found that the temperature of normal healing wounds would be around 37.8 °C, whereas an increase or decrease in temperature of 2.2 °C might lead to deterioration of the injured area. A reduction in temperature indicates an ischemic condition, while an increase in temperature indicates infection and inflammatory responses. As a result, temperature monitoring has a lot of potential as a good way to assess wound status. Temperature sensors based on different detection mechanisms are infrared sensors, thermistors, and resistance temperature sensors. Some of the components used for sensor preparations are nanofibers/nano meshes of polyurethane, graphene, platinum, and gold [27,28].

### 3.3. Pressure Responsive Wound Dressings

Individuals with diabetic foot ulcers and pressure ulcers often undergo therapies to relieve pressure. This pressure on the injured site is caused by friction, vigorous movements, shear, and external pressure itself, which leads to blockage of blood circulation, resulting in the death of surrounding cells, and tissues and impeding the healing process. Also, patients with immobility, especially those who are bedridden, are more prone to these conditions. The emergence of pressure sensor dressings made it easy to monitor wounded areas. There are four types of sensors depending on their working principles: piezoresistive, capacitive, triboelectric, and piezoelectric sensors. Development of multiple flexible pressure sensors for wound monitoring is under way [29,30,31,32]. 

### 3.4. Moisture Responsive Wound Dressings

The moisture level in the wound area is crucial in all phases. An excessive amount of moisture may lead to macerated tissues, and less moisture hinders healing by desiccating the wound surfaces. High moisture content in the wound may be due to exudates from the injury itself, excessive sweating, incontinence in urination, or trans epidermal water loss. Excessive wound moisture leads to uncountable removal of dressings and may also cause damage to surrounding areas of skin, resulting in maceration. Thus, dressings with real-time sensors would be more advantageous than regular dressings [33,34]. In 2016, the first commercially available moisture-sensitive wound dressing called “Wound Sense” was introduced [35]. The manufacturing of breathable dressings with sensor incorporation is made using various composite materials (polyvinyl alcohol, carbon nanotubes, graphene oxide, graphene, nanosheets made of palladium and cerium oxide) [28].

### 3.5. Sustained Drug Releasing Wound Dressings

Stimuli-responsive dressings, which are externally regulated, allow monitoring and control of the release of drugs to the injured sites [36]. Sustained drug release can be attained by incorporating drugs into dressing layers, in which hydrogels are extensively studied and employed. These drug delivery dressings are made of polymers such as poly lactide-co-glycolide, polyvinyl pyrrolidone, polyvinyl alcohol, poly hydroxyalkyl-methacrylates, polyurethane-foam, hydrocolloid, alginate, hyaluronic acid, collagen, and chitosan [37,38]. 

Wound dressings are also exhibit inbuilt self-healing properties. Some dressings are come in injectable forms, and a few come in sprayable forms. Multiple dressing forms were prepared by incorporating multi-functional aspects to treat various ranges and stages of healing. In the current review, we have endeavored to collate the most recent research data on multifunctional wound dressing materials as well as investigations on their advantages in wound healing.

## 4. Articles Search for Narrative Review-Inclusion & Exclusion Criteria

This narrative review and electronic search were conducted by assessing all studies available on the PubMed, Scopus, and Google Scholar databases from Jan 2020 to Jan 2022. The phrase for the search of various dressings included the following: “Hydrogel” AND “wound” AND “dressing”, “Film” AND “wound” AND “dressing”, “Foam” AND “wound” AND “dressing”, “Sponge” AND “wound” AND “dressing”, “Nanofiber” AND “wound” AND “dressing”, “Gauze” AND “wound” AND “dressing”, “plant extract” AND “wound” AND “dressing”. Combining results from all databases, the articles obtained for the year 2021 were 62,781 and 50,835 articles for the year 2020. With further screening being done by removing duplicates and excluding review articles, letters, meta-analyses, systemic reviews, conference papers, short communications, case studies, and languages other than English, the final number was brought down to 3315 for 2021 and 2794 for 2020 (Figure 3). 

After the primary screening, the available full-length articles were screened to obtain relevant studies about specified dressings which included animal experimentation. In this review, a total of 45 different studies were included, in which 22% is hydrogel dressings (*n* = 10), foam includes 7% (*n* = 3), sponge comprises of 18% (*n* = 8), films include 13% (*n* = 6), nanofibers comprise of 13% (*n* = 6), gauzes and plant extracts hold 9% (*n* = 4) and 18% (*n* = 8) respectively. 

## 5. Wound Dressing

In 2500 BC, Mesopotamians were using clay tablets to treat various ailments and people cleaned the injuries with milk or water prior to actually applying the resin and honey dressing. In 460–370 BC, Hippocrates of ancient Greece were using wine or vinegar to clean wound surfaces. Romans introduced four fundamental inflammation concepts (rubor/redness, tumor/swelling, calor/heat and dolor/pain). Antiseptic techniques advanced significantly after the introduction of antibiotics to control infections. Modern wound dressings were developed in the twentieth century. Currently, over 5000 wound care products are available [39,40,41]. 

The most prevalent wound dressings come in various structures and shapes, including foams, hydrogel, topical formulations (herbal extracts, enzymatic formulations, ointments), gauze, films, nanofibers/composites, etc. (Figure 4). 

### 5.1. Hydrogels

Hydrogels are polymeric substances extensively employed in drug release, cell culture, the epidermis, blood vessels, and muscular and smooth tissue engineering. These are 3D networks made of hydrophilic polymer bonds formed by mechanical or covalent crosslinking using physical or chemical processes [42]. Four prominent hydrogel crosslinking strategies are ionic interactions, crystallization, complementary group chemical reactions, and radical polymerization [43,44,45]. Hydrogels are moist, non-particulate, nontoxic, and non-adherent, which are all foremost properties if the wound bed has to be kept pain-free, well-hydrated, and well-oxygenated [46]. 

Many hydrogels that lack reliable flexibility could be deformed or disrupted by mechanical pressure because of being implemented in the injured area, which could lead to cracks in the hydrogels, jeopardizing the dressing’s integrity as a physical shield and letting bacteria enter the wound surface. Therefore, self-repairing dressing components with antimicrobial activities are highly valued [47]. Sabzevari et al. designed hydrogel in combination with anti-microbial peptide human cathelicidin P-18/LL-37 derived from conditioned medium of genetically modified Wharton’s jelly derived-mesenchymal stem cells. This hydrogel were shown to exhibit higher regenerative potential and wound repair [48]. Likewise, RRP9W4N peptides [49], (naphthalene-2-ly)-acetyl-diphenylalanine-dilysine-OH (NapFFεKεKOH) [50], Bmkn2 [51] were also incorporated with hydrogels to hinder the growth of infectious micro-organisms thus accelerating wound healing. Studies of composite hydrogels fabricated with antibacterial agent nisin & EDTA and coagulating agent fibrinogen, incorporation of nuclear factor erythroid 2–related factor 2 (NRF2) shown to have excellent ability of epithelialization and wound recovery [52,53]. 

Contemporarily designed dressings in adhesive, sprayable or injectable forms possess biological safety, antioxidant, antibacterial, self-healing properties, stimuli responsive, and preserve moistness in the micro ambient while enhancing regeneration by impacting the restorative phases of injury [54,55,56,57]. Likewise, they render an excellent forum to pack cells, antimicrobial drugs, supplementary factors for growth and various biomolecules of interest (Table 1) [58,59]. Hydrogels are extensively employed dressings but with few drawbacks such as low mechanical strength, multifunctional/smart dressing may be expensive, biocompatibility issues might arise in synthetic hydrogels, may cause skin maceration, may be incompatible with excessively exudate wounds.

### 5.2. Films

Bloom et al. first documented the adoption of modern plastic film as a dressing in 1945, describing how he used cellophane to treat burns on 55 prisoners during World War II, which influenced the work of Bull et al., who soon developed a transparent film dressing made of nylon [69,70]. Presently, transparent film dressings are polymer membranes of varying thicknesses that are adhesive-coated on one side. The polyurethane layer is moist and gas permeable, reducing the risk of tissue maceration by preventing moisture accumulation in the wound, and its transparency allows monitoring of the injured area [71]. Films are used as a primary or secondary dressing to treat partial-thickness wounds with little or no exudate, necrosis, and infection. They are available in many sizes, both sterile and bulk films, which are light, elastic, and quickly adhere to injuries with intricate shapes and curves. Film dressings have scope for use on several wound types, for example, post-operative wound healing by primary intention, superficial burns, and skin grafts, as they allow for easy wound monitoring [72,73,74].

Many studies have shown that films also serve as platforms for loading various biomolecules, drugs, and growth factors (Table 2). Arruda et al. loaded xyloglucan films with Concanavalin A, a protein that activates the immune system, Rezvanian et al. proved the effectiveness of simvastatin-loaded films using diabetic wound models, Kausar et al. and Mahmood et al. incorporated antibacterial agents such as vancomycin and ofloxacin respectively, epidermal growth factor was loaded into Gelatin films by Tanaka et al. which shown to encourage the proliferation of keratinocytes and fibroblasts [75,76]. 

The most significant advantage of films is that they are transparent, allowing physicians to supervise injuries without removing wound dressing, lowering the infection risk, trauma, and suffering during dressing changes. This, however, may be less attractive to that same patient, who might prefer not to see the wound. Film dressings are inappropriate for highly wet injuries or hemostatic application, and later removal of films might cause pain and epidermal damage in some cases [71,77].

**Table 2 pharmaceutics-14-01574-t002:** Summary of recent advances in film dressings.

Dressing Composition	Dressing Material Evaluated/ Group of Subjects	Key Findings	Reference
Sodium alginate and pectin loaded with simvastatin (SIM)	Control, Saline, Hydrogel film, Kaltostat^®^ commercial dressing, SIM-hydrogel film	Better angiogenic effect contributed accelerated healing, quicker re-epithelialization and improved collagen deposition	[78]
*Hammada scoparia* leaf extract (PSP) and poly (vinyl alcohol) (PVA)	Saline, Cytol centella cream, PVA film (100%), (70:30) PSP/PVA film	Hastened wound closure and reepithelialisation	[79]
Xyloglucan dressing (XG) and Concanavalin A ©	Saline, XG, XGC	Non-toxic, homogenous, angiogenesis, remodelling, early epithelialization	[80]
Chitosan film (CF) loaded with Vancomycin (V)	Saline, Burn, Burn vancomycin, Burn + CF, Burn + VCF2	Controlled drug release, remarkable antimicrobial effect and enhanced wound recovery	[81]
Polycaprolactone (PCL), Gelatin, poly (perfluoro decyl methacrylate) + poly (dimethyl siloxane) + poly (perfluoro decyl methacrylate (PMFA)	PCL-Gelatin, PCL-PMFA, PCL G + F	Non-adherent ability and constant drainage performance	[82]
Ofloxacin (O), tea tree (T) and lavender oil (L) in gellan gum hydrogel film	Blank, O, L, T, OL, OT	Antibacterial and wound-healing properties are notable	[83]

### 5.3. Sponges

High adhesive properties are a must-have criterion for stopping bleeding. Because the sponge structure possesses adhesive characteristics, the implant surface adheres to the wounded organ’s surface without utilizing additional suture material or other techniques [84]. In addition to adhesive qualities, other significant aspects such as absorbability and sorption are based on the chemical structure and spatial structural organization. The determination of efficiency is done by the method used to create the morphological basis of the samples, which is usually a sponge structure of animal origins such as collagen or synthetic sources such as cellulose salts [84,85].

In 2019, Hartinger et al. developed a vancomycin-releasing hemostatic sponge made of collagen (derived from *Cyprinus carpio*) and evaluated its efficacy in infected incision wound models of Wistar rats. They observed a statistically significant release of vancomycin; colony-forming units were lower with vancomycin-loaded sponges compared to the placebo group [86]. Zhao et al. developed a multifunctional 10% Kangfuxin interlinked carboxymethyl chitosan/alginate sponge, which has proven to assist in faster wound closure than other groups such as 5% & 15% treated groups [87]. A porous nanocomposite sponge studied by Rongxiu et al., comprising graphene oxide, polyvinyl alcohol, and sodium alginate, had excellent absorbability, antimicrobial effects, and cytocompatibility and can be used to treat wounds with more exudates [88]. Sponges loaded with antimicrobials such as penicillin, streptomycin, and amoxicillin were prepared to study their hemostatic, coagulating, antimicrobial, and faster healing rates, respectively [89,90]. Many significant studies (Table 3) conducted using newly synthesized multifunctional sponge dressings were found to display hemostatic effects, biocompatibility, reduced bioburden, and reduced wound closure time. In some cases, sponge dressings might be mechanically unstable; may cause maceration due to higher moisture content; and, in the absence of antibiotics, may lead to the development of microbial infections. They are also not suitable for dry wounds like secondary burn wounds.

### 5.4. Nanofibers/Nanocomposite

Electro-spun nanofibers are a novel type of material with varying fiber sizes in the nanometer range that is created through a number of techniques including template synthesis, phase separation, drawing, self-assembly, and electrospinning. Because of its ease of production, roughly comparable easiness over the procedure, and ease of scale-up, electrospinning appears to be among the most compelling of these strategies. These properties aid in cell recognition, ECM structure similarity, and enhanced protein binding, all of which result in better biocompatibility. Due to their high surface-area-to-volume ratio, excellent porous structure, and plasticity, they are employed in medical utilizations, including scaffolds for regenerative medicine, delivering therapeutic agents, and as dressings for various wounds [99,100]. Nanofibrous forms of various polymers have been introduced as an artificial extracellular matrix (ECM) [101]. Chitosan [102], collagen [103], gelatin [104] and silk are the most cited natural polymers used as electro spun nanofibrous scaffolds, while polylactic acid [105], poly-lactic-co-glycolic acid [106], polyglycolic acid, polycaprolactone [104] and poly-caprolactone/lactide copolymer are the most widely employed synthetic polymers.

Electrospinning wound dressings with nanofibers have several advantages. For starters, the structural and physiological properties are comparable to those of the natural ECM [107,108]. Furthermore, the electrospinning polymer matrix can integrate the biostability of natural polymeric substances with the uniform automated potency of artificial polymeric substances [101]. The porosity structure of the nanofiber membrane allows for the effective loading of a variety of physiologically active components, including antibacterial medicines, nanostructured materials, vitamins, growth hormones, and herbal extracts (Table 4) [109,110]. By adjusting the structure and size of fibers, the release of biological molecules or therapeutic agents can be regulated, thus enabling the effectual recovery of the injured area. Therefore, electro-spun nanofibrous materials exhibit significant possibility in the production of modern bioactive wound dressings [99,111]. There is a need for the development of nanofibrous dressings to be compatible with wounds with heavy bleeding, high exudate formation, and lower production costs.

### 5.5. Foams

In foams, a semi-obstructive outer layer surrounds the polyurethane or silicone core. The hydrophobic external surface repels fluids and germs while still facilitating oxygenation [115,116]. They are potential wound drainage absorbents; therefore, the frequency of dressing change is low. In addition, since foam is non-adhesive, in the course of changing, the foam dressings may reduce injury and adjoining dermal bruise The hydrofiber, which turns into a gel when it comes into contact with a wet wound, is made with a polyurethane layer, and the hydrofiber layer, in turn, has a layer of foamed polyurethane or polyurethane with a filmy texture. Foam dressings come in various thicknesses and can be adherent or nonadherent. The foams are frequently provided with a film backing that serves as a moisture and microbe-resilient shield to the environment. The conductivity of the film backings varies, influencing the efficiency of water vaporization and gaseous exchange. All of these are considered foam dressings [116,117].

The foam dressing designed by Miaomiao et al. with multi-layers of polyvinyl alcohol, sodium carboxymethylcellulose mesh, and drug Stearyl trimethyl ammonium chloride displayed dynamic exudate absorption, antimicrobial effect, and coagulation of blood [118]. Several types of foam dressings contain silver (Ag), which is active in reducing microbial load and possibly speeding up wound healing (Table 5). Foam dressings maintain a moist environment in and around the wounded area, promote good absorption, are hemostatic, and provide good adherence. Yet foams may tightly adhere to a wound if the wound dries. They are opaque in nature, the gaseous exchange is limited, and excessive exudate absorption by foams might lead to maceration of surrounding skin.

### 5.6. Gauzes

Gauze is an ancient dressing used by the Egyptians to wrap bodies before burial. Gauze products are of two subcategories based on fabric construction or material composition they are (i) woven and (ii) non-woven. Non-woven gauze dressings are made of rayon or synthetic fibers and are developed to replace woven products because they adhere less to the wound bed and produce less lint. Woven products, also known as absorbent gauze, are typically made of 100 percent natural cotton yarns, which are at use over centuries. Cutting these woven gauzes will result in shed of fibers and is susceptible to linting with fibers [122]. Gauzes that dry quickly have traditionally been used to cover and treat damage in the dermal area. Dermal coverings that produce and maintain balanced moisture content are considered best for wound healing [74]. Traditional gauze is likely inexpensive, compatible, readily available, and frequently used in surgical and clinical practices. Recently developed gauze materials are modified with polymers, nanoparticles, and other components to make them more reliable for use in various wound therapies. They lack qualities such as being dry and lacking moisture balance, which may disrupt the healing wound and cause tissue damage when removed, and frequent replacement is required if saturated with wound exudates (Table 6).

### 5.7. Others

Currently, more attention being paid on dressings prepared from natural and synthetic sources [126]. Plant extracts [127,128,129], Adult stem cell therapies [130,131,132] studied effectively with injectable forms and loaded on to dressings were also found to be effective in skin regeneration. Numerous biochemical parameters, including cell proliferation and immune function, are regulated by proteolytic enzymes. Exogenous protease was the very first enzymatic treatment used to treat chronic wounds [133]. Biopolymers and bioactive compounds used in wound dressings are obtained from natural sources: plants, animals, and even microbial [134,135]. For manufacturing dressings based on wound type and requirements, biopolymers such as chitosan, cellulose, hyaluronic acid, alginate, elastin, dextran, fibrin, pectin, gelatin, collagen, and fibronectin are most often used [136]. A number of plant (papain, ficin, actinidin) and bacterial proteases or peptide (collagenase) based formulations are under development to target chronic wound debridement, possessing antimicrobial, angiogenesis, hemostatic, anti-inflammatory, anti-scarring properties (Table 7) [137].

### 5.8. Plant Derived Bioactive Compounds and Biopolymers in Wound Therapy

Ayurvedic preparations, biomolecules, drugs, and pharmaceutically important plants have minimal side effects, as well as less side effects than synthetic drugs, according to research on traditional and medicinal plants. Medicinal herbs and herbal preparations for tissue repair are inexpensive. Polyphenols, alkaloids, flavonoids, steroids, tannins, and terpenoids are just a few of the phytoconstituents derived from medicinal plants. Honey is one such kind of solution produced by bees using nectar from plants, and it is rich in water, monosaccharides, disaccharides, proteins, amino acids, minerals, and pigments (Figure 5). Many research studies have proven that honey is effective at all stages of the wound healing process. They have excellent anti-inflammatory and antibacterial effects [143,144]. Curcumin, demethoxycurcumin, and bisdemethoxycurcumin are three major phytoconstituents sourced from *Curcuma longa*. These components are proven to kill viruses and bacteria, having anti-oxidative and anti-inflammation properties [145,146]. Aloe vera is a plant that grows in hot climate regions and arid regions. This plant contains major derivatives belonging to the group of anthracenes along with sugars, enzymes, minerals, vitamins, and pigments. The remarkable healing potential of Aloe vera is due to the presence of glucomannan, which regulates fibroblast growth factors, collagen synthesis, and secretion [147,148,149]. Nimbolinin, nimbin, nimbidin, nimbidol, sodium nimbinate, gedunin, salannin, quercetin, nimbanene, 6-desacetylnimbinene, nimbandiol, nimbolide, and nimbiol are extracted from various parts of *Azadirachta indica*. Quercitin has a vital role as an anti-microbial agent and an anti-inflammatory agent [150,151,152].

### 5.9. Polymers

Cellulose is a polysaccharide predominately employed for manufacturing scaffolds, matrices, dressings for chronic wounds, and shortening healing time. Cellulose is synthesized by bacteria such as *Acetobacter xylinum* and plants. The triggering effect on granulative and epithelializing phases in partial and full-thickness wound models proves that it has the potential to accelerate wound healing without any side effects. It is used in regenerative medicine as a wound-healing scaffold for severely damaged skin and small-diameter blood vessel replacement due to its similarity to ECM. Cellulose is an innovative product that is recommended as an alternative dressing for superficial partial-thickness burn wounds because it is a natural biomimetic, biodegradable, antibacterial, skin—friendly, and non-toxic polysaccharide [153].

Chitin is a simple and abundant -(14) glycan made up of 2-acetoamido-2-deoxy-d-glucopyranose units. It is a key element of arthropod shells such as crabs, shrimp, lobsters, and insects, and it is also formed in extracellular environment by fungi and some brown algae. Chitin is a highly water-insoluble compound that is a byproduct or waste of the crab, shrimp, and crawfish processing industries. Chitosan is a useable and basic sequential polysaccharide derived from the N-deacetylation of chitin in the presence of alkali. Chitin and chitosan have been shown to have antitumor, hypocholesterolemic, and antihypertensive properties. As chitosan is present in abundance, also non-toxic and being biocompatible, development of dressings using chitosan is given importance. On the other hand it is also active against bacterial and fungal infections [154]. Collagen is the most abundant, triple-helical, and structural protein present in human beings. Collagen is produced by fibroblast cells, which play an important role in all phases of wound healing. Because of its degradability, biocompatibility, and ability to promote the organization and accumulation of new collagen, it has numerous biomedical applications. Despite its excellent biological performance, collagen’s poor mechanical properties, high degradability, and inability to prevent the growth of bacteria have limited its biomedical applications. As a result, cross-linkers and other materials are frequently used in the manufacturing of collagen-based dressings. Gelatin, as a derivative, is similar to collagen, and due to its excellent biocompatibility and biodegradability, it can also be used as a raw material for wound dressing [155].

Hyaluronic acid is a biopolymer that relates to the glycosaminoglycan family of heteropolysaccharides found in the human vitreous humour, joints, rooster comb, umbilical cord, epidermis, and connective tissue. Furthermore, HA can be obtained through bacterial fermentation. As a major component of the edema fluid, HA promotes the recruiting process of neutrophil cells, which are engaged in the phagocytosis of debris and the removal of dead tissue, as well as the subsequent release of tumour necrosis factor-alpha (TNF-), interleukin-1 (IL-1) and interleukin-8 [156]. Silk is a naturally existing polymer with numerous applications in medicine, particularly wound dressing. Silk fibres are derived from silkworms and are primarily composed of proteins, fibroin, and sericin, which are made up of eighteen different amino acids. Silk is used in a variety of biomedical applications, including healing process, cytocompatibility, blood compatibility, high tensile strength, and oxygen permeability. Silk fibroin is an excellent treatment for chronic wounds, diabetic foot ulcers, and burns. Because of its biocompatible and mechanical properties, it is combined with other biomaterials such as alginate, elastin, silver sulfadiazine, and epidermal growth factors in the form of films, hydrogels, and electrospun nanofibers. The silk-blend dressings promote keratinocyte proliferation and migration, have a greater affinity for fibroblasts, and influence ECM secretion, thereby improving wound healing [157].

## 6. Conclusions

In wound healing, the employment of biomaterials and bioactive chemicals dates back to ancient times, but synthetic materials and nanoparticles have proven to be essential in developing a successful treatment plan. Most films, foams, hydrogels, sponges, and composite dressings are exploited in clinical settings as wound healing products are composed of biomaterials. Multifunctional dressings’ investigation focuses primarily on those that encapsulate bioactive substances, sustain drug release, are stimuli sensitive, homeostatic, and have plenty of other prospects for clinical benefits. Still, some of these products do not release biologicals that directly enhance the healing process. These studies can be considered viable to some extent. Despite the emergence of multiple trials with abundant treatments, there is a lack of high-quality evidence. They are still in the initial phases and therefore do not meet all of the demands of modern evidence-based treatments due to the small number of animals, varying sizes and degrees of wounds, the timeframe of observation, and restricted methods of monitoring wound healing. Unrelenting attempts to create, if not a universal, then at least an optimal drug acting at all stages of wound healing and using the maximum number of natural mechanisms of tissue regeneration have not yet been crowned with success. Conducting more trials would determine the efficacy of these dressings in clinical settings. The core objective would be to imitate the aspects of fetal tissue regeneration in the mature healing process, involving whole hair and glandular restoration without postponement or scarring. Emerging treatments based on biomaterials, nanoparticles, and proteases with biomimetic properties hold much potential for improving wound care and will be a valuable addition to the therapeutic toolbox for treating slow-healing wounds.

## Figures and Tables

**Figure 1 pharmaceutics-14-01574-f001:**
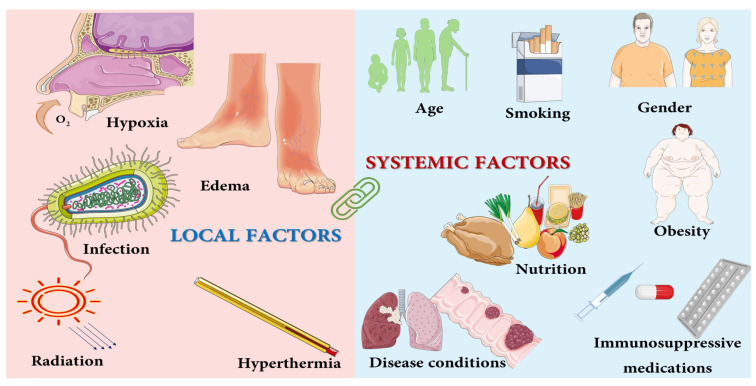
Factors affecting wound healing.

**Figure 2 pharmaceutics-14-01574-f002:**
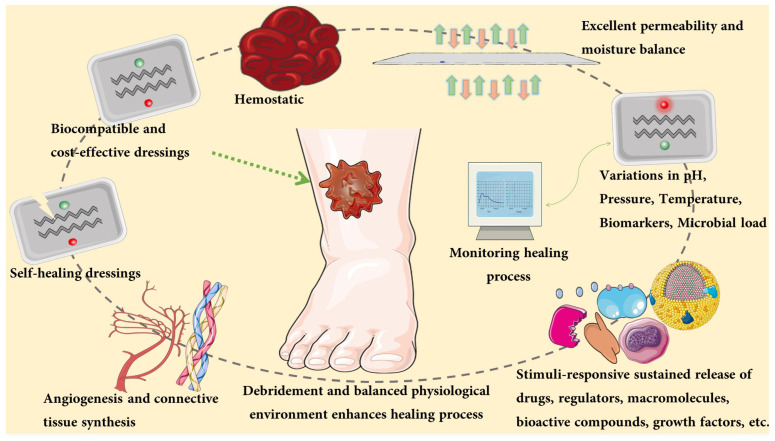
Characteristic features of multifunctional smart wound dressings.

**Figure 3 pharmaceutics-14-01574-f003:**
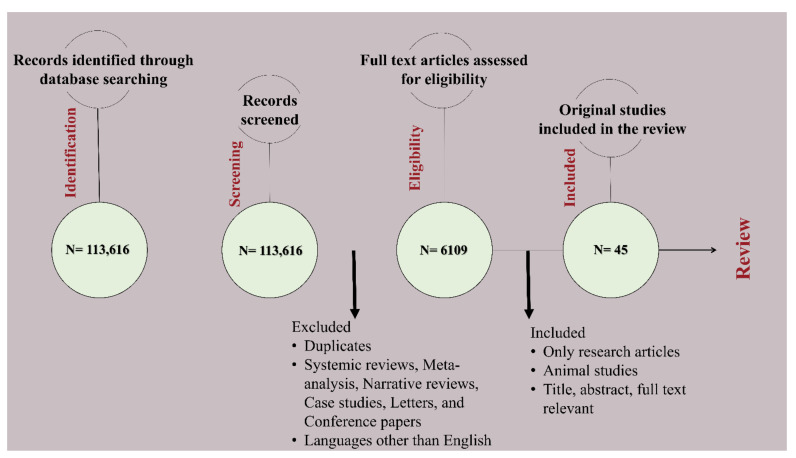
Flow chart of article selection and review process.

**Figure 4 pharmaceutics-14-01574-f004:**
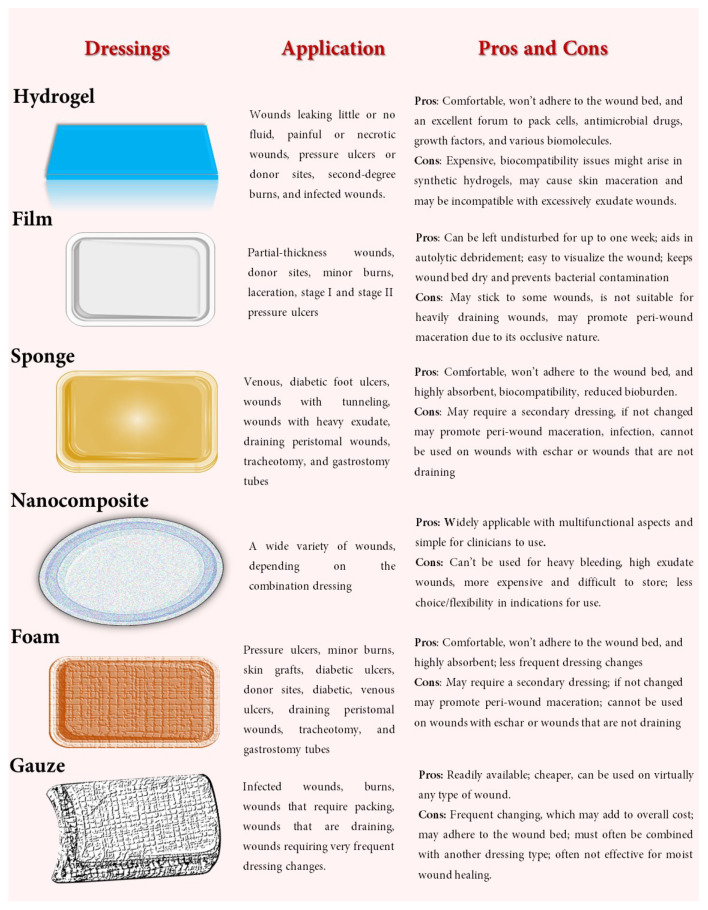
Types of dressings, applications, pros and cons.

**Figure 5 pharmaceutics-14-01574-f005:**
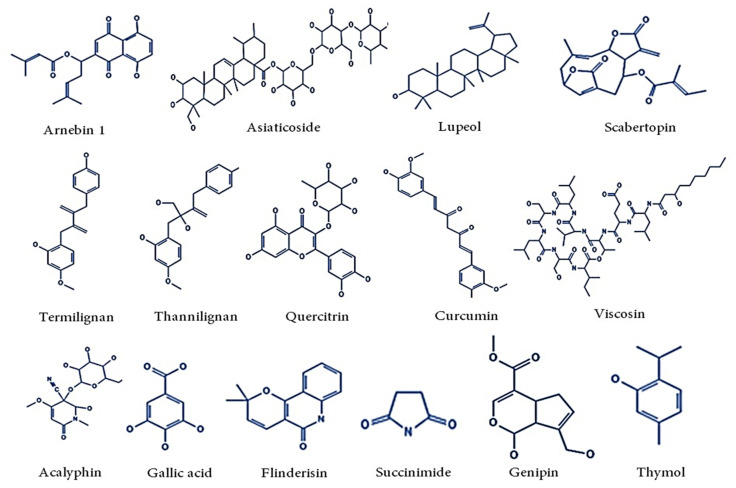
Phytochemicals from various plant sources that have been found to be effective in wound treatments.

**Table 1 pharmaceutics-14-01574-t001:** Summary of recent advances in hydrogel dressings.

Dressing Composition	Dressing Material Evaluated/ Group of Subjects	Key Findings	Reference
Gelatin methacrylate (GM), methacrylate of silk fibroin (MSF), stem cells (SC) and platelet concentrate plasma (PCP)	Gauze, GM/MSF, GM/MSF/PCP, GM/MSF/PCP/SC	Wound healing, reepithelialisation, and collagen deposition are all accelerated.	[60]
N-carboxyethyl chitosan, hyaluronic acid–aldehyde, insulin and adipic acid dihydrazide	Control, hydrogel, hydrogel + insulin	Expedited re-epithelialization and neovascularization; shortened inflammatory phase; increased granulation tissue formation; facilitated collagen deposition	[61]
Polymerized ionic liquid (PL), konjac glucomannan (K) and electrical stimulation (ES)	Control, PL-K-0, PL-K-5, PL-K-10, PL-K-20, ES, PL-K-ES-20	It has great mechanical qualities and biocompatibility, and it disinfect quickly and effectively	[62]
Poly [2-(methacryloyloxy) ethyl] dimethyl-(3-sulfopropyl) ammonium hydroxide (SBMA), quaternized chitosan methacrylate (QCS) and Gelatin methacrylate (GelMA)	Control, SQG hydrogel	Improves granulation tissue development and collagen deposition by demonstrating good cell activity, hemocompatibility, and histocompatibility	[63]
Cannabidiol (CD), Zinc (Zn^2+^) ions and the alginate polymer (Alg)	Control group, Tegaderm™ group (3M), Alg@Zn group, CD/Alg@Zn group	Controlling of inflammatory infiltration, collagen deposition and granulation tissue production, and blood vessel formation	[64]
Gelatin meth acryloyl (GMa), Cerium oxide nanoparticles (CeNs) and an antimicrobial peptide (AMP)	GMa-Dopa, GMa-Dopa-AMP, GMa-Dopa-CeNs, GMa-Dopa-AMP-CeNs	Spray ability, adhesiveness, antibacterial activity, as well as the ability to scavenge ROS and regenerating skin are all promising	[57]
Gelatin (G), methacrylic anhydride (M), sodium tetraborate and oxidized dextran (BD)	GelMA/OD/Borax hydrogel	Efficiently stopped the bleeding, decreasing bioburden and hastened the healing of the wound	[65]
	blank group, positive group (PBS) and G-M-BD		
	G-M-BD-L prepolymer solution, G-M-BD prepolymer solution		
Rose Bengal (RB), graphene oxide (GO), polyvinyl Alcohol (PVA) hybrid hydrogel and chitosan microspheres	PVA, GO/PVA, β-GO/PVA, and β-GO/RB/PVA HDs	Biocompatibility and water-absorbing capability are desirable, as is an antimicrobial impact	[66]
N-(9-fluorenylmethoxy carbonyl)-L-phenylalanine (Fmoc-F) and berberine chloride (BBR)	Control, Fmoc-F/BBR, Fmoc-F/BBR + Light	Antibacterial and anti-biofilm action has been improved	[67]
Chitosan hydrogel membrane (CS), Cerium oxide nanoparticles (CeO_2_) from *Abelmoschus esculentus* extract	CS, CS-1% CeO_2_, CS-5% CeO_2_	Showed excellent microbicidal, antioxidant activity and proven to accelerate healing time and closure	[68]

**Table 3 pharmaceutics-14-01574-t003:** Summary of recent advances in sponge dressings.

Dressing Composition	Dressing Material Evaluated/ Group of Subjects	Key Findings	Reference
Chitosan, alginate, hyaluronic acid, genipin	Medical gauze, CAHS1, CAHS2 and CAHS3	Facilitates wound closure and haemostatic	[91]
Kang Fuxin (K), Alginate (AG) and carboxymethyl chitosan (CMC)	Control, AC, AG, ACK-5, ACK-10, ACK-15	Good moisture transmission, plasticity, antimicrobial features, low cytotoxicity, and haemostatic	[87]
Chain based sponge dressing (CSD)	CSD and gauze	Achieved haemostasis quickly	[92]
Janus polyurethane, Superhydrophobic silica nanoparticles, super hydrophilic polyurethane (PU)	Saline, PU Sponge, Janus PU sponge	Reduced the risk of infection, excessive wetting and accelerated the efficiency of healing	[93]
Platelet rich plasma (PRP), collagen sponge scaffold (CSS) with modified polydopamine (PDA)	PDA-CSS-PRP, CSS-PRP, CSS and Normal Saline	Fast angiogenesis, rapid collagen arrangement leading to less scar development	[94]
Corn stalk (CS), silver nanoparticles (AgNPs) and chitin sponge (CH)	Control, polyvinyl formal sponge, CH-CS-AgNPs, CH-CS, CH	Biocompatibility and nontoxicity, fast wound closure rate	[95]
Chitosan and H. syriacus petroleum ether extract (SPC)	Control, Standard Mebo^®^, SPC-plain, SPC-low, SPC-medium, SPC-high	Perfect re-epithelization and epidermal remodelling	[96]
recombinant collagen (rCOL)	Implantation test with rCOL, COL	Perfect biocompatibility with no sensitivity, no toxicity, no stimulation reactions observed and excellent haemostatic effect	[97]
	Saline, rCOL sponge extraction		
	Saline, Saline extraction, Solvent saline, Seasame oil Solvent extraction, non-polar solvent extraction		
*Andrias davidianus* dermal secretion (ADDS)- nanocrystals of cellulose and nanofibers of cellulose (CS) sponge	Control, Gelatin sponge, ADDS-CS3, ADDS-CS2	Excellent haemostatic properties	[98]

**Table 4 pharmaceutics-14-01574-t004:** Summary of recent advances in nanofiber dressings.

Dressing Composition	Dressing Material Evaluated/ Group of Subjects	Key Findings	Reference
Antimicrobial peptide KRWWKWWRRC (K), Collagen nanosheet (CN)	Blank, CN, KCN,	Good reepithelialisation, rapid wound closure and low inflammation	[103]
Poly (caprolactone) (PCL), Poly (vinyl alcohol) (PVA), collagen nanofibers (Col), *Momordica charantia* pulp extract (Ex)	Control, PCL/PVA/Col, PCL/PVA/Col/Ex 1%, PCL/PVA/Col/Ex 5%, PCL/PVA/Col/Ex 10%	Hemocompatible, cytocompatibility, and prevent bacterial penetration	[104]
Nanofiber mat of cellulose acetate (CA) loaded with parathyroid hormone related protein (Pthrp-2)	CA, CAP-1%, CAP-5%	Promoted epithelialization, collagen deposition and blood vessel formation	[110]
*Moringa oleifera* seed (MOS) polysaccharide (PS), nanocomposite with silver (AgNPs)	Distilled water, betadine ointment, MOS-PS-AgNPs-25, MOS-PS-AgNPs-50, MOS-PS-AgNPs-100	Supports wound tightening and tissue generation as well	[112]
*Salvia officinalis* extract-assisted biosynthesis route to synthesize zinc oxide and Magnetite-based nanocomposites	mupirocin^®^ ointment, magnetite ointment, zinc oxide/magnetite ointment, control	Granulation tissue, collagen density and epithelization improvements observed	[113]
Bilayer scaffold consists curcumin dextran nanoparticles (CDN), cerium oxide nanoparticles (CON) loaded Gelatin cryogel layer and polyvinyl alcohol-poly (vinylpyrrolidone)-iodine-potassium iodide layer (GCL)	Control, GCL, GCL-CDN-CON, Tegaderm pad commercial	Strong antioxidant, antibacterial and faster wound closure	[114]

**Table 5 pharmaceutics-14-01574-t005:** Summary of recent advances in foam dressings.

Dressing Composition	Dressing Material Evaluated/ Group of Subjects	Key Findings	Reference
Mesostructured cellular silica foams (MCF) decorated with silver ions (Ag)	Control, MCF and MCF-Ag	MCF-Ag antibacterial haemostatic agent with splendid water absorption and antibacterial capacity	[119]
Polyurethane biomacromolecule combined foam (PUC), asiaticoside (AS), Silver nanoparticle (AgNPs)	PUC-AS-AgNPs, commercial gauze	shorter wound closure time, higher reepithelialisation and less pain score	[120]
Hyaluronic acid, a cell wall fragment of *Cutibacterium acnes* (LimpiAD)	LimpiAD foam	Prevented skin lesions or any sign of skin damage	[121]

**Table 6 pharmaceutics-14-01574-t006:** Summary of recent advances in gauze dressings.

Dressing Composition	Dressing Material Evaluated/ Group of Subjects	Key Findings	Reference
Gauze (G), Quat 188, silver nanoparticles (AgNPs), oxytetracyline hydrochloride (Ox) and salicyl-imine-chitosan biopolymer (SIC)	Gauze fabric, Dermazin Ointment, G/Ag NPs/Ox, G/Ag NPs/Ox/CS, fabrics/Ag NPs/Ox/SIC-0.2, G/Ag NPs/Ox/SIC-0.4, G/Ag NPs/Ox/SIC-0.6	Promoting fibrosis and collagen reorganization	[25]
Cotton guaze (CG), chitosan (C), Gelatin (G) and alginate (A)	Control, CG and AGCCg-5	High fluid absorption, excellent biocompatibility, hemocompatibility, haemostatic performance, low cost, reliability, safety, and a simple manufacturing process	[123]
Amino-modified cotton gauze (CG), poly (carboxybetaine-co-dopamine methacrylamide) (PCM) copolymer, silver nanoparticles (AgNPs)	Blank, Pristine CG, PCM@AgNPs-CG	Showed excellent hemocompatibility, cytocompatibility, reduced the inflammatory response and wound infection	[124]
Gauze, polydopamine, perfluorocarbon and silver nanoparticle (Lotus@Gauze)	Vaseline^®^ petrolatum gauze, Atrauman^®^Ag gauze, Lotus@Gauze, Irradiated Vaseline^®^ petrolatum gauze, Irradiated atrauman^®^Ag gauze, Irradiated Lotus@Gauze	Antiadhesive and antibacterial gauzes	[125]

**Table 7 pharmaceutics-14-01574-t007:** Summary of recent studies in plant extracts and proteases.

**Dressing Composition**	**Dressing Material Evaluated/** **Group of Subjects**	**Key Findings**	**Reference**
*Bergenia ciliata* rhizome ethanolic extract	Control, Povidine ointment, *Bergenia ciliata*, 5% (*w*/*w*) and 10% (*w*/*w*) ointments	Wound healing is faster and wound contraction is better.	[128]
*Bridelia micrantha* methanol leaf extract	Silver sulphadiazine cream, aqueous cream, 10% BME aqueous cream, 2.5% BME aqueous cream, 0.625% BME aqueous cream	Angiogenesis, collagenation, and re-epithelization all improved, as did antibacterial and antioxidant activities	[127]
Serine protease (Tricuspidin) from *Tricosanthus tricuspidata*	Tricuspidin & Trypsin	Excellent proteolytic ability, anti-inflammatory effect	[133]
	Tricuspidin & PBS		
*Plantago major* extract (PM), ursolic acid (UA) and oleanolic acid (OA)	Gel, Gel (Mebo), PM, UA, and OA gels	Non-toxic & improve wound healing	[138]
*Urtica dioica* extract, Chitosan (C), gold (G)/perlite nanocomposite ointment(P)	Control, mupirocin^®^ ointment, P, PG, PGC	Decreasing the length of healing time and stimulates MRSA-infected wound regeneration	[139]
Cysteine protease (Drupin) from *Ficus drupacea* (Fd) latex	Saline, Neosporin, papain, Fd-protein rich fraction, drupin, drupin-IAA	Controlled expression of MMP’s, increased collagen production, cellular migration and proliferation	[140]
Bromelain based Escghar ex (ESX)	ESX group, Gel arm group	Good debridement activity of the formulation	[141]
Serine protease (SP), Antiquorin (Aq) *Euphorbia antiquorum*	Saline, Aq, SP + Aq	Improved platelet aggregation and quick haemostatis	[142]

## Data Availability

The data that support the findings of this study are available in standard research databases such as PubMed, Science Direct, or Google Scholar, and/or on public domains that can be searched with either key words or DOI numbers.

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
