# Peer review of "Multifunctional and Smart Wound Dressings—A Review on Recent Research Advancements in Skin Regenerative Medicine"

_pharmaceutics, 2022, doi:10.3390/pharmaceutics14081574_

Round 1

Reviewer 1 Report

1.     In this manuscript, the authors summarized the ideal characteristic and different forms of wound dressings. However, they failed to highlight the theme of “multifunctional” and “smart”, only simply listed advantages of each form and servals studies in corresponding section.

2.     The authors mentioned that ideal wound dressing should be non-adhesive, however such statement is controversial. To date, a large number of adhesive wound dressings are reported. Especially under certain scenarios, adhesivity are essential.

3.     Natural polymers and bioactive chemicals are referred several times throughout the manuscript, however related content is lacking. The application and advantages of natural-derived substance may be added in the review.

4.     The given figures are labeled incorrectly, two “Figure 2” are presented while “Figure 1” is missing. Besides, some are unnecessary in this review. (For example, the wound healing process is wildly summarized in a plethora of studies. However, such process is not the main point of this manuscript that only mentioned in the introduction part and no connections with the following content. Similarly, the Figure “factors affecting wound healing” is too simple to present as a separate figure.) Instead, figures focused on the dressing forms, functionalization strategies, smart response principles should be more favorable for this manuscript.

5.     Likewise, the contents in tables are way to redundant. Column 2-5 can be omitted while more description of the materials should be supplemented.

Author Response

All the authors in unison thank you for all the valuable comments and suggestions which have helped us to present the information in a much more lucid manner.

Reviewer 2 Report

The paper is a review of the main wound dressings currently used in the medical practice. The authors have produced a valuable work with numerous tables and figures that make the entire paper more organized and accessible for the reader. I have however 3 major observations:

1. The authors have to specify at the beginning of the paper, after the introduction, the period they investigated in terms of publications, which databases they took into consideration, what inclusion/exclusion criteria they applied and how many results their investigations revised. In addition, a graphical presentation of this process should be introduced.

2. Throughout the text, the authors used very small sentences, displaying an almost telegram style. The text should be re-written in my opinion by re-organizing the sections and sentences in longer, more readable phrases.

3. The section "Discussions" is useless in a review since the entire paper is basically a discussion of others' papers; it can be combined with the conclusions.

Author Response

(The authors gave the same response as above.)

Reviewer 3 Report

In the manuscript entitled "Multifunctional and Smart Wound Dressings - A Review on Recent Research Advancements in Skin Regenerative Medicine", the authors summerized the recent advances in skin wound dressings including hydrogels, films, sponges, nanofibers, foams, gauzes, and thers. This review provides veluable introductions of wound dressings, especially for beginners. It will be better if the authors can discuss more of the current issues, perspectives, and outlook. English style and syntax need to be improved, please check the full manuscript.

Author Response

(The authors gave the same response as above.)

Round 2

Reviewer 1 Report

I think that the authors appropriately answered all of the questions by the reviewers. And the revision was appropriately done and the revised manuscript is much improved.

Reviewer 2 Report

Accept